# Screening for Antibacterial Activity of French Mushrooms against Pathogenic and Multidrug Resistant Bacteria

**Clément Huguet [1], Mélanie Bourjot [1], Jean-Michel Bellanger [2], Gilles Prévost [3] and Aurélie Urbain [1,*]**

1 Faculty of Pharmacie, Institut Pluridisciplinaire Hubert Curien, Université de Strasbourg, CNRS, UMR 7178, CAMBAP, 67081 Strasbourg, France; huguet@unistra.fr (C.H.); bourjot@unistra.fr (M.B.)
2 CEFE, CNRS, Université Montpellier, EPHE, IRD, INSERM, 34090 Montpellier, France; jean-michel.bellanger@cefe.cnrs.fr
3 UR 7290, Institut de Bactériologie, Université de Strasbourg, 67081 Strasbourg, France; prevost@unistra.fr
* Correspondence: urbain@unistra.fr

**Abstract:** In the alarming context of antibiotic resistance, we explored the antibacterial potential of French mushrooms against wild-type and multidrug-resistant (MDR) bacteria. In order to accelerate the discovery of promising compounds, screenings were carried out by TLC-direct bioautography. A total of 70 extracts from 31 mushroom species were evaluated against five wild-type bacteria: *Staphylococcus epidermidis*, *Staphylococcus aureus*, *Enterococcus faecalis*, *Escherichia coli*, and *Pseudomonas aeruginosa*. This first screening revealed that 95% of the extracts contained antibacterial compounds. Generally, it was observed that EtOAc extracts exhibited more active compounds than methanolic extracts. In addition, all extracts were overall more active against Gram-positive bacteria than against Gram-negative strains. The most promising mushroom extracts were then screened against various multidrug-resistant strains of *S. aureus* and *E. coli*. Activity was globally less on MDR strains; however, two mushroom species, *Fomitopsis pinicola* and *Scleroderma citrinum*, still contained several compounds inhibiting the growth of these MDR pathogenic bacteria. Stearic acid was identified as a ubiquitous compound contributing to the antibacterial defence of mushrooms. This screening revealed the potential of macromycetes as a source of antibacterial compounds; further assays are necessary to consider fungal compounds as promising drugs to counter antibiotic resistance.

**Keywords:** antibiotic resistance; mushrooms; antibacterial; screening; direct bioautography; thin-layer chromatography

## 1. Introduction

Antibiotic resistance has become a major public health concern. According to the Centers for Disease Control and Prevention (CDC), infections due to multidrug-resistant (MDR) bacteria are responsible for one death every 15 min in the US [1], and the most recent analysis estimates that in 2019 nearly 5 million deaths worldwide were associated with bacterial resistance, including 1.27 million deaths directly attributable to MDR bacteria [2]. *Escherichia coli* and *Staphylococcus aureus* are the two main pathogenic bacteria involved in deaths associated with drug resistance. The World Health Organization (WHO) estimates that the number of deaths due to antibiotic resistance could even reach 10 million by 2050 if no action is taken, including the development of new effective therapies [3]. The overwhelming majority of current antibiotics are derived from microorganisms, but surprisingly little attention has been paid to alternative resources such as macromycetes. Only recently, pleuromutilin derivatives from Basidiomycete fungi have been approved for human use in 2019 by the Food and Drug Administration (FDA). Yet mushrooms represent a remarkable and original reservoir of antibacterial compounds that remains to be investigated deeper [4].

To unveil promising fungal species in the fight against pathogenic bacteria, we have evaluated the antibacterial potential of 70 mushroom extracts from 31 species growing

in North East France. In order to accelerate the identification of promising species, we have implemented thin-layer chromatography-direct bioautography (TLC-DB). All extracts were evaluated against five pathogenic wild-type bacteria, including *Staphylococcus aureus*, *Escherichia coli* and *Pseudomonas aeruginosa*, listed as priority pathogens by the WHO, and for which there is an urgent need to identify new antibiotics [3]. The most promising mushroom extracts were subsequently evaluated by TLC-DB towards vour MDR strains of *S. aureus* and vour MDR strains of *E. coli*, obtained from clinical isolates. These screenings revealed that various fungal compounds inhibited the growth of MDR bacteria. Two mushroom species, the polypore *Fomitopsis pinicola* and the common earthball *Scleroderma citrinum*, were found to be the most interesting fungi regarding the overall number and selectivity of active compounds. Stearic acid has been identified as a common constituent contributing greatly to the antimicrobial activity of mushroom extracts.

## 2. Materials and Methods

### 2.1. Mushrooms

Fruiting bodies of 30 different wild mushroom species were collected from their natural habitats in the vicinity of Strasbourg, France, between 2014 and 2019. Three species were harvested multiple times, from different places or different years: *Neoboletus erythropus*, *Morchella esculenta*, and *Scleroderma citrinum*. Two authors, M. Bourjot and C. Huguet, taxonomically identified sporocarps on the basis of micro- and macroscopic characteristics, and comparison with literature [5,6]. The identification of the specimens was confirmed by DNA analyses by J.-M. Bellanger. The edible common mushroom *Agaricus bisporus* added to the screening was purchased from a local supermarket. Information about the species collected is provided in Table 1. For each harvest, a voucher specimen was freeze-dried prior to deposit at the Laboratory of Analytical Chemistry and Pharmacognosy, University of Strasbourg.

**Table 1.** Information about the mushrooms evaluated in the present study (authors of mushroom scientific names have been omitted for the sake of conciseness).

| Order | Family | Species | Harvest Information * |
|---|---|---|---|
| Agaricales | Agaricaceae | *Agaricus bisporus* | 2016 February, from supermarket |
| | | *Macrolepiota procera* | 2015 October, Fuschloch |
| | Amanitaceae | *Amanita citrina* | 2014 October, Haguenau |
| | | *Amanita muscaria* | 2015 October, Dabo |
| | Bolbitiaceae | *Bolbitius variicolor* | 2016 April, Illkirch |
| | Cortinariaceae | *Cortinarius semisanguineus* | 2015 November, Wisches |
| | Mycenaceae | *Mycena rosea* | 2017 Octpber, Ohlungen |
| | Omphalotaceae | *Gymnopus luxurians* | 2016 June, Illkirch |
| | Physalacriaceae | *Armillaria cepistipes* var. *pseudobulbosa* | 2015 October, Illkirch |
| | | *Armillaria ostoyae* | 2018 October, Floessplatz |
| | Pleurotaceae | *Pleurotus ostreatus* | 2018 November, Barr |
| | Pluteaceae | *Volvopluteus gloiocephalus* | 2016 April, Illkirch |
| | Psathyrellaceae | *Parasola auricoma* | 2016 June, Illkirch |
| | Strophariaceae | *Hypholoma fasciculare* | 2015 October, Fuschloch |
| | Tricholomataceae | *Clitocybe nebularis* | 2017 October, Ried |
| Boletales | Boletaceae | *Neoboletus erythropus* #1 | 2018 October, La Fischhutte |
| | | *Neoboletus erythropus* #2 | 2017 August, Saint-Jean-Saverne |
| | Hygrophoropsidaceae | *Hygrophoropsis auriantaca* | 2016 October, Hohbuhl |
| | Paxillaceae | *Paxillus involutus* | 2015 October, Dabo |
| | Sclerodermataceae | *Scleroderma citrinum* #1 | 2018 October, Balbronn |
| | | *Scleroderma citrinum* #2 | 2018 October, Floessplatz |
| | | *Scleroderma citrinum* #3 | 2018 October, Floessplatz |
| | | *Scleroderma citrinum* #4 | 2019 September, Balbronn |

**Table 1.** *Cont.*

| Order | Family | Species | Harvest Information * |
|---|---|---|---|
| Pezizales | Morchellaceae | *Morchella esculenta* #1 | 2019 April, Erstein |
| | | *Morchella esculenta* #2 | 2019 April, Plobsheim |
| | Pezizaceae | *Peziza vesiculosa* | 2016 April, Illkirch |
| Polyporales | Fomitopsidaceae | *Fomitopsis pinicola* | 2014 October, Barr |
| | Ganodermataceae | *Ganoderma applanatum* | 2018 October, Balbronn |
| | Meripilaceae | *Meripilus giganteus* | 2017 August, Saint-Jean-Saverne |
| | Polyporaceae | *Polyporus umbellatus* | 2017 July, Still |
| | | *Trametes versicolor* | 2018 August, Saint-Jean-Saverne |
| Russulales | Albatrellaceae | *Scutiger pes-caprae* | 2018 April, Fackental |
| | Russulaceae | *Lactarius helvus* | 2017 September, Haguenau |
| | | *Lactifluus piperatus* | 2017 July, Still |
| | | *Russula integra* | 2017 August, Still |
| | | *Russula lepida* | 2017 August, Saint-Jean-Saverne |

* Date and location in Alsace region, France.

### 2.2. Chemicals

Heptane, ethyl acetate (EtOAc) and methanol (MeOH) used for extractions were of technical grade (Carlo Erba, Val-de-Reuil, France). Analytical grade dichloromethane ($CH_2Cl_2$) (Merck, St Louis, CA, USA), methanol (MeOH) (Carlo Erba, Val-de-Reuil, France), methyl tert-butyl ether (MTBE), tetrahydrofuran (THF) (Fluka Chemicals, Buchs, Switzerland), and cyclohexane (Fisher Scientific, Loughborough, UK) were used for TLC separation. The vital dye reagent 3-(4,5-dimethylthiazol-2-yl)-2,5-diphenyltetrazolium bromide (MTT) used for the visualization of antibacterial activity was purchased from Merck (Damstadt, Germany). Gentamicin (VWR, Sanborn, New York, NY, USA) and polymyxin B (Pfizer, Paris, France) were used as positive controls.

### 2.3. Extracts Preparation

Mushroom fruiting bodies were frozen at −18 °C and then freeze-dried using a Labconco Freezone 4.5 L freeze dryer. Dried mushrooms were then grinded into a thin powder with a Retsch ZM 200 grinder. The resulting powder was extracted successively with three solvents of increasing polarity—heptane, EtOAc, and MeOH—under the same conditions: for each solvent, the dry matter was extracted twice in a row by maceration under magnetic stirring at room temperature during 24 h, with a ratio of 50 mL of solvent per gram of dry matter. After filtration on Whatman filter paper, final extracts were evaporated under reduced pressure to remove solvent and stored at 4 °C for further assays.

### 2.4. Thin-Layer Chromatography

Thin-layer chromatography was performed on normal particle size silica gel 60 F254 plates (Merck, Darmstadt, Germany) 10 × 20 cm. Each precoated TLC Si60 F254 aluminum-backed plate was formerly washed in an elution chamber (CAMAG, Geneva, Switzerland) with MeOH. After thorough drying, 100 µg of mushroom extracts were deposited as lines of 8 mm length on the TLC plate using a Linomat 5 CAMAG, and development was performed in a twin trough glass chamber over a distance of 8 cm. Two different solvent systems were used according to the polarity of the extracts: for the EtOAc extracts, MTBE-THF-cyclohexane (5:1:4 $v/v/v$) was used as mobile phase, whereas $CH_2Cl_2$-MeOH (9:1 $v/v$) was used for the methanolic extracts. After TLC separation of mushroom extracts, the TLC plates were placed under chemical hood overnight to remove traces of residual solvent.

### 2.5. Bacterial Culture

#### 2.5.1. Wild-Type Bacteria

*Staphylococcus aureus* (ATCC 29213), *Enterococcus faecalis* (ATCC 29212), *Escherichia coli* (ATCC 25922), and *Pseudomonas aeruginosa* (UCBPP PA14) were obtained from our in-house

collection. The strain of *Staphylococcus epidermidis* (ATCC 12228) was kindly given by Dr Didier Blaha from University Claude Bernard Lyon 1, France.

The culture of the bacterial strains and the bioautography process were carried out in a biosafety Class II cabinet (Thermo Fisher, Lagenselbold, Germany). All the bacterial strains were cultivated on Mueller–Hinton (MH) supplemented with agar (VWR chemicals, Leuven, Belgium) between screenings. Colonies of each strain were streaked using the quadrant technique and sub-cultured every 24 h. For the screening purposes, the bacterial strains were cultivated in MH broth, except *E. faecalis* which was cultivated in Brain Heart Infusion (BHI) broth (Becton Dickinson, Le Pont-de-Claix, France). Prior to bioassays, an isolated bacterial colony of each strain was taken from the solid culture, put into 10 mL of suitable broth in a sterile Erlenmeyer, and incubated at 37 °C for 20 h on a shaker incubator at 120 rpm in a bacterial culture chamber (Edmund bühler Gmbh Th15, Bodelshausen, Germany). Then, 1 mL of this stock bacterial suspension was diluted into 20 mL of fresh MH or BHI broth. This new bacterial suspension was incubated at 37 °C on a shaker incubator at 120 rpm until it reached an optical density at the wavelength of 600 nm ($OD_{600}$) between 0.4 and 0.5 (measured on a UV-2401 PC spectrophotometer from Shimadzu, Kyoto, Japan), corresponding to the exponential bacteria growth phase.

### 2.5.2. Multidrug Resistant Bacterial Strains

The four multidrug-resistant strains of *S. aureus* (named SA1, SA2, SA6, and SA8) and the four MDR strains of *E. coli* (EC2, EC5, EC6, EC8) were isolated from patients from the Strasbourg University Hospitals. The SA1, SA2, and SA6 strains came from blood cultures and the SA8 strain from auricular pus. The four MDR strains of *E. coli* came from cytobacteriological urine tests. The antibiograms of the resistant *S. aureus* were performed by disc diffusion method. The susceptibility towards antibiotics of the four *E. coli* MDR bacteria was evaluated using an automated dilution method by Vitek2 (Biomérieux, Craponne, France). The resistance phenotype of the different MDR bacterial strains is illustrated in Tables A1 and A2 and takes into account the recommendations of the Antibiogram Committee of the French Society for Microbiology and the European Committee on Antimicrobial Susceptibility Testing (EUCAST), with respect to the sensitivity to the different antibiotics tested [7]. The MDR bacterial strains were cultivated and prepared under the same conditions applied to the wild-type bacterial strains described above.

### 2.6. TLC-Direct Bioautography

After elution and drying of the TLC plates, a positive control (30 µg) was applied on each plate, as a vertical band to be unambiguously distinguished from fungal samples. Gentamicin was mainly used, except for TLC plates with *E. coli* MDR strains resistant to gentamicin, for which we used polymyxin B. Bacterial suspensions were sprayed with an airbrush (Airbrush Compressor ARP150, Ningbo, China) until the TLC plates were homogeneously wet. Plates were subsequently put into square sterile Petri dishes saturated with water and incubated at 37 °C for 5 h. After this, the TLC plates were sprayed with a 5 mg/mL MTT aqueous solution and incubated at 37 °C overnight. After reduction of the yellow MTT into blue-purple formazan, antibacterial activity was detected as pale inhibition zones contrasting against the colored background. Visualization and photography of the bioautograms were performed with a Reprostar 3 (Camag, Muttenz, Switzerland).

### 2.7. Purification and Identification of Stearic Acid

The EtOAc extract of *Lactifluus piperatus* was partitioned by centrifugal partition chromatography (CPC) using a Sanki (Tokyo, Japan) LLB-M apparatus equipped with a column of 230 mL rotating at 1200 rpm. The extract was partitioned in ascending mode with the solvent system heptane-EtOAc-MeOH-$H_2O$ (5:5:6:4 *v/v/v/v*) at a flow rate of 5 mL/min, using Gilson 322 pumps (Middleton, WI, USA). The elution was recorded with a Gilson Ultraviolet/Vis-151 detector, and fractions were collected every 30 s with a Gilson FC 204 automatic collector. Fractions containing the compound of interest were pooled to

undergo a final purification step by preparative HPLC through a Kinetex column C18 100 Å 100 × 21.2 mm, 5 μm, Axia packed (Phenomenex, Torrance, CA, USA). The structure of stearic acid was confirmed by comparison with a standard (Sigma-Aldrich, St. Louis, MO, USA) by direct-infusion mass spectrometry and nuclear magnetic resonance spectroscopy (micrOTOF-Q II and NMR Spectrometer Avance III 400 MHz respectively, both Bruker, Karlsruhe, Germany).

## 3. Results

### 3.1. Mushroom Collection

A total of 30 wild mushroom species were collected in their natural habitats in Alsace, northeastern France, between 2014 and 2019. This region consists of the Upper Rhine plain in the east and the Vosges mountains in the west, offering varied biotopes. The harvests were carried out in all seasons, in various environments: forests, meadows, roadsides, etc. Several species were collected several times, in different places, and/or different periods. Attention was paid to select mushrooms not damaged or attacked, neither too young nor too old. In addition, to these wild mushrooms, we have decided to include to the screening the common mushroom *Agaricus bisporus*, supplied from a local supermarket and grown in a cave on a specific compost. The final collection comprised fifteen Agaricales species, five Polyporales, five Russulales, four Boletales, and two Pezizales species (Table 1).

### 3.2. Antibacterial Activities of Mushroom Extracts against Wild-Type Bacteria

Seventy macromycetes extracts (35 EtOAc extracts and 35 MeOH extracts) were evaluated for their antibacterial potential against three Gram-positive bacteria: *Enterococcus faecalis*, *Staphylococcus aureus*, *Staphylococcus epidermidis*, and against two Gram-negative bacteria: *Escherichia coli* and *Pseudomonas aeruginosa*. The bacteria *S. aureus*, *E. coli*, and *P. aeruginosa* have been chosen as they are listed as priority pathogens according to the WHO. Although *E. faecalis* and *S. epidermidis* are not included in this list, these species are subject to resistance increasing at a non-negligible rate, and which exhibit resistance phenotypes similar to those listed for priority pathogens by the WHO. Our hypothesis that mushrooms produce antibacterial compounds to adapt to their competitive environment was verified, as all mushroom samples exhibited at least one compound active against one bacteria. On bioautograms, active compounds appeared as pale yellow bands, as MTT was not reduced by living bacteria to a blue-purple derivative. The first global observation is that Gram-positive bacteria were much more susceptible to fungal extracts than Gram-negative bacteria. In addition, *E. faecalis* was generally more affected than *Staphylococcus* species. This global observation is illustrated in Figure 1, with the effect of an extract from *Armillaria cepistipes* var. *pseudobulbosa* against the five reference strains. While at least twelve inhibition bands could be clearly detected against *E. faecalis*, a significant decrease in the number of active compounds was observed on the bioautograms against the other bacteria. Just one antibacterial band with a retention factor (Rf) of 0.75 was present on the bioautogram with *E. coli*, while barely any inhibition band could be clearly distinguished against *P. aeruginosa*.

This first example also illustrates the main advantage of TLC bioautography over microdilution assays, as this approach is much more informative regarding the constituents involved in antimicrobial activity. Indeed, diffusion and dilution methods give an idea of the antibacterial potential of an extract, but cannot assess the number nor first clues about the compounds involved in this bioactivity. In addition, TLC bioautography enables us to notice active compounds specific to a fungal species or specific against a bacterial strain.

This is also exemplified in Figure 2, which presents bioautograms against the three Gram-positive bacteria obtained for EtOAc extracts from five different mushrooms. We can observe on all bioautograms an inhibition zone at Rf values around 0.78 for each of the five fungal species, indicating the presence of a common low-polarity compound that inhibits the growth of the three tested Gram-positive bacteria. Bioautography also enables us to reveal active compounds specific to a fungal species. This is illustrated with an inhibition band at Rf 0.65 present only in *Scleroderma citrinum* #2 (lane 3), active

against the three bacterial strains. In addition, Figure 2 shows that some compounds are more selective; for example, mushrooms 1 to 4 possess a compound eluting at Rf 0.45 that prevents the growth of *E. faecalis*, but that has no effect on both *Staphylococcus*. Similarly, two compounds from *Fomitopsis pinicola* (lane 5) eluting at Rf 0.20–0.30 inhibit the growth of *E. faecalis* and *S. aureus*, but do not affect the development of *S. epidermidis*. The polypore *Fomitopsis pinicola* also contains more polar compounds (Rf 0–0.15) that inhibit the growth of both *Staphylococcus* but not the growth of *E. faecalis*. These examples demonstrate the diversity and selectivity of antibacterial compounds present in mushrooms.

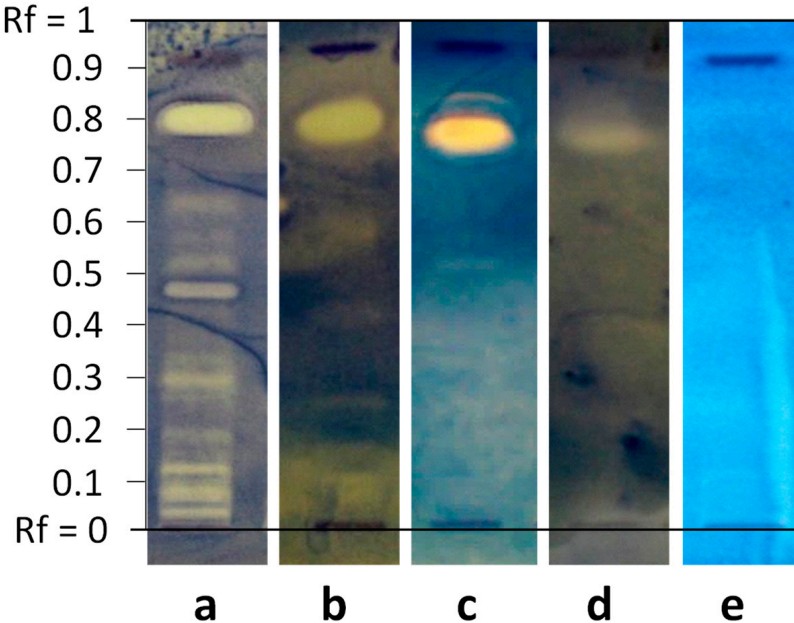

**Figure 1.** TLC bioautograms obtained with the EtOAc extract from *Armillaria cepistipes* var. *pseudobulbosa* against (**a**) *E. faecalis*; (**b**) *S. aureus*; (**c**) *S. epidermidis*; (**d**) *E. coli*; (**e**) *P. aeruginosa*.

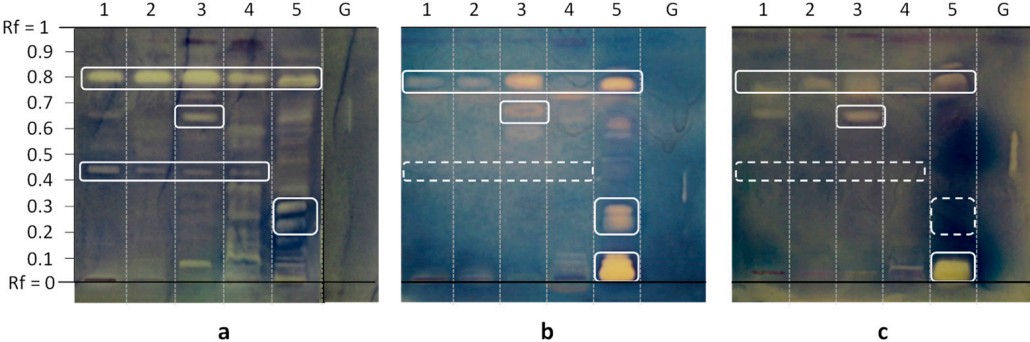

**Figure 2.** TLC bioautograms obtained with EtOAc extracts from 1: *Neoboletus erythropus* #1; 2: *Peziza vesiculosa*; 3: *Scleroderma citrinum* #2; 4: *Amanita muscaria*; 5: *Fomitopsis pinicola*; G: gentamicin, against (**a**) *E. faecalis*; (**b**) *S. aureus*; (**c**) *S. epidermidis*.

The overall number of inhibition zones is reported in Table 2, and details of inhibition zones with corresponding retention factors (Rf) values can be found in Tables A3–A7.

**Table 2.** Number of inhibition zones induced by mushrooms extracts on bioautograms (the higher the number of zones, the darker the background). *Ef*: *Enterococcus faecalis*, *Sa*: *Staphylococcus aureus*, *Se*: *Staphylococcus epidermidis*, *Ec*: *Escherichia coli*, *Pa*: *Pseudomonas aeruginosa*.

| Order | Mushroom Species | EtOAc Extracts | | | | | MeOH Extracts | | | | |
|---|---|---|---|---|---|---|---|---|---|---|---|
| | | *Ef* | *Sa* | *Se* | *Ec* | *Pa* | *Ef* | *Sa* | *Se* | *Ec* | *Pa* |
| Agaricales | *Agaricus bisporus* | 2 | 3 | 2 | 3 | 0 | 5 | 1 | 1 | 2 | 0 |
| | *Amanita citrina* | 4 | 3 | 3 | 3 | 1 | 2 | 1 | 1 | 2 | 0 |
| | *Amanita muscaria* | 8 | 7 | 2 | 2 | 1 | 2 | 2 | 2 | 1 | 0 |
| | *Armillaria cepistipes* var. *pseudobulbosa* | 12 | 5 | 5 | 1 | 0 | 1 | 1 | 1 | 0 | 0 |
| | *Armillaria ostoyae* | 1 | 4 | 2 | 3 | 0 | 0 | 0 | 0 | 0 | 0 |
| | *Bolbitius variicolor* | 2 | 1 | 2 | 1 | 0 | 3 | 1 | 2 | 1 | 0 |
| | *Clitocybe nebularis* | 2 | 4 | 7 | 7 | 2 | 0 | 0 | 0 | 0 | 0 |
| | *Cortinarius semisanguineus* | 1 | 3 | 3 | 0 | 0 | 0 | 0 | 2 | 0 | 0 |
| | *Gymnopus luxurians* | 4 | 0 | 0 | 1 | 1 | 0 | 0 | 2 | 0 | 0 |
| | *Hypholoma fasciculare* | 4 | 2 | 3 | 2 | 1 | 5 | 3 | 1 | 0 | 4 |
| | *Macrolepiota procera* | 5 | 2 | 0 | 0 | 0 | 0 | 0 | 2 | 0 | 0 |
| | *Mycena rosea* | 4 | 4 | 4 | 2 | 2 | 0 | 0 | 0 | 0 | 0 |
| | *Parasola auricoma* | 4 | 5 | 2 | 2 | 1 | 0 | 0 | 1 | 0 | 0 |
| | *Pleurotus ostreatus* | 2 | 4 | 6 | 3 | 1 | 0 | 0 | 1 | 0 | 0 |
| | *Volvopluteus gloiocephalus* | 11 | 6 | 3 | 3 | 2 | 0 | 0 | 1 | 0 | 0 |
| Boletales | *Hygrophoropsis aurantiaca* | 5 | 1 | 2 | 2 | 0 | 1 | 0 | 2 | 0 | 0 |
| | *Neoboletus erythropus* #1 | 6 | 2 | 2 | 2 | 0 | 0 | 0 | 0 | 0 | 0 |
| | *Neoboletus erythropus* #2 | 4 | 6 | 6 | 1 | 2 | 0 | 3 | 1 | 0 | 0 |
| | *Paxillus involutus* | 9 | 3 | 1 | 1 | 1 | 0 | 0 | 1 | 0 | 0 |
| | *Scleroderma citrinum* #1 | 9 | 6 | 3 | 2 | 4 | 0 | 0 | 1 | 0 | 0 |
| | *Scleroderma citrinum* #2 | 7 | 6 | 2 | 2 | 2 | 0 | 0 | 1 | 0 | 0 |
| | *Scleroderma citrinum* #3 | 4 | 4 | 1 | 3 | 0 | 0 | 5 | 1 | 1 | 0 |
| Pezizales | *Morchella esculenta* #1 | 1 | 2 | 1 | 0 | 1 | 1 | 4 | 0 | 1 | 0 |
| | *Morchella esculenta* #2 | 4 | 0 | 4 | 2 | 2 | 0 | 1 | 0 | 0 | 0 |
| | *Peziza vesiculosa* | 5 | 2 | 1 | 1 | 0 | 2 | 4 | 2 | 1 | 2 |
| Polyporales | *Fomitopsis pinicola* | 14 | 10 | 4 | 5 | 3 | 5 | 1 | 8 | 3 | 3 |
| | *Ganoderma applanatum* | 4 | 3 | 1 | 1 | 0 | 1 | 4 | 1 | 2 | 0 |
| | *Meripilus giganteus* | 7 | 4 | 2 | 2 | 0 | 0 | 0 | 2 | 1 | 0 |
| | *Polyporus umbellatus* | 8 | 3 | 4 | 3 | 0 | 2 | 1 | 2 | 0 | 0 |
| | *Trametes versicolor* | 2 | 4 | 4 | 2 | 0 | 0 | 1 | 1 | 0 | 0 |
| Russulales | *Lactarius helvus* | 4 | 6 | 3 | 6 | 2 | 0 | 1 | 1 | 1 | 0 |
| | *Lactifluus piperatus* | 3 | 2 | 3 | 2 | 1 | 2 | 0 | 0 | 0 | 0 |
| | *Russula integra* | 7 | 1 | 3 | 1 | 1 | 2 | 0 | 2 | 0 | 0 |
| | *Russula lepida* | 5 | 8 | 1 | 5 | 2 | 0 | 0 | 2 | 0 | 0 |
| | *Scutiger pes-caprae* | 7 | 2 | 3 | 1 | 0 | 1 | 1 | 1 | 1 | 0 |

Results from this initial screening on wild-type bacteria have orientated the selection of mushroom extracts to be evaluated against multidrug-resistant strains. We have selected mushrooms according to (i) their antibacterial activity profile, including number of active compounds but also selectivity and effect against Gram-negative bacteria, (ii) the available biomass necessary to further purify active compounds, and (iii) the lack of knowledge regarding mycochemistry, which could increase chances to identify novel bioactive chem-

ical structures. Based on these considerations, six fungal species have been selected for evaluation on MDR strains:

- *Fomitopsis pinicola* (Polyporales) was selected because both polar and intermediate-polarity extracts were shown to contain several molecules affecting the growth of the five wild-type bacteria.
- The sulphur tuft (*Hypholoma fasciculare,* Agaricales) was selected because its MeOH extract was one of the few active against *P. aeruginosa*, with at least four antibacterial compounds.
- *Clitocybe nebularis* (Agaricales) was included in the bioassay as its EtOAc extract was one of the most active against *E. coli*, as assessed by the total number of inhibition bands.
- The oyster mushroom (*Pleurotus ostreatus*, Agaricales) was included for further investigation because of the activity profile of its EtOAc extract against Gram-negative bacteria, and above all to its availability as an edible mushroom.
- Regarding the common earthball (*Scleroderma citrinum*, Boletales), the EtOAc extract of the first specimen (#1) was one of the most active in terms of number of inhibition zones against all bacteria. In addition to the three initial samples screened against reference bacteria, a fourth sample harvested in 2019 was included in the screening against MDR strains.
- Finally, the peppery milkcap (*Lactifluus piperatus*, Russulales)—that has been used for the extraction of stearic acid—was also included in the bioautography assays against MDR strains.

### 3.3. Antibacterial Activity against MDR Bacterial Strain

Eighteen fungal extracts have been evaluated by TLC-direct bioautography against eight clinically isolated strains resistant to conventional drugs. These eight strains, four *S. aureus* and four *E. coli*, can be considered as good models of MDR bacteria, since they display distinct antibiograms (Tables A1 and A2), and for some of them a high level of resistance. For instance, the *E. coli* isolate EC2 was found to be resistant to 16 antibiotics out of 28 tested. Gentamicin was used as positive control for bioautograms with *S. aureus* as the four strains were susceptible to this broad spectrum aminoglycoside antibiotic; on the contrary, the four MDR *E. coli* strains were resistant to gentamicin, and therefore, polymyxin B was used instead.

#### 3.3.1. Antibacterial Activity against MDR *S. aureus* Strains

Except SA1 that was susceptible to oxacillin (a narrow-spectrum antibiotic closely-related to methicillin), the majority of *S. aureus* strains evaluated in this study were MRSA, i.e., methicillin-resistant *S. aureus*, responsible for infections associated with higher health-care costs and mortality rate, due to resistance to beta-lactam antibiotics [8]. The search for effective compounds against MRSA remains a great challenge. The clinical isolate SA6 was the one with the highest resistance in this panel, with proven resistance to six antibiotics belonging to various classes such as penicillins, aminoglycosides, fluoroquinolones, and fusidanes (Table A1). The Table 3 reports the Rf values of all inhibition bands visible on bioautograms against the different MDR *S. aureus* strains.

Generally, fewer bands were observed compared to the reference strain of *Staphylococcus aureus*, and MeOH extracts affected less the growth of MDR strains than EtOAc extracts, which is consistent with the results from the preliminary screening on wild-type bacteria. Methanolic extracts from Agaricales species did not have any effect on the growth of these MDR strains.

Conversely, both low-polarity and polar extracts from *Fomitopsis pinicola* were found to contain several antibacterial compounds affecting all bacterial strains. Interestingly, some differences of selectivity could be observed. As seen in Figure 3, bioautograms are slightly different according to the *S. aureus* strain. While some compounds inhibited only the wild strain (Rf between 0.20–0.30, zone **e**), others inhibited all strains, such as the compound **a** at Rf 0.75. More surprisingly, some inhibition bands were observed only against MDR strains, like the band **b** present in the four bioautograms with clinical isolates, or the bands **c** and **d**

inhibiting SA2, SA6, and SA8. An inhibition band around Rf 0.15 was found to be specific to SA2, and was not even present in the bioautogram of the reference strain of *S. aureus* (**f**). Therefore, these results indicate that several compounds from the widespread brown-rot mushroom *F. pinicola* can prevent the development of MRSA strains.

**Table 3.** Rf values of inhibition bands induced by mushrooms extracts on bioautograms against the four MDR *S. aureus* strains SA1, SA2, SA6, and SA8.

| Mushroom Species | Extract | SA1 | SA2 | SA6 | SA8 |
|---|---|---|---|---|---|
| *Clitocybe nebularis* | EtOAc | 0.75 | 0.25, 0.45, 0.50, 0.75 | 0.75 | 0.40, 0.75 |
| | MeOH | / | / | / | / |
| *Hypholoma fasciculare* | EtOAc | 0.01, 0.75 | 0.01, 0.75 | 0.75 | 0.01, 0.75 |
| | MeOH | / | 0.08 | / | 0.08 |
| *Pleurotus ostreatus* | EtOAc | 0.75 | 0.25, 0.65, 0.75 | 0.40, 0.65, 0.75 | 0.10, 0.20, 0.65, 0.75 |
| | MeOH | / | / | / | / |
| *Lactifluus piperatus* | EtOAc | 0.75 | 0.75 | 0.75 | 0.01, 0.75 |
| | MeOH | / | 0.70 | 0.05, 0.70 | 0.70 |
| *Fomitopsis pinicola* | EtOAc | 0.01, 0.10, 0.35, 0.50, 0.70, 0.75, 0.85 | 0.01, 0.05, 0.08, 0.10, 0.18, 0.40, 0.55, 0.70, 0.75 | 0.01, 0.45, 0.62, 0.70, 0.75 | 0.01, 0.08, 0.40, 0.55, 0.65, 0.70, 0.75 |
| | MeOH | 0.55, 0.65, 0.70, 0.80 | 0.60, 0.70, 0.78, 0.85 | 0.60, 0.70, 0.78, 0.85 | 0.50, 0.65, 0.70, 0.85 |
| *Scleroderma citrinum* #1 | EtOAc | 0.08, 0.50, 0.60, 0.70 | 0.08, 0.55, 0.65, 0.75 | 0.08, 0.60, 0.70, 0.75 | 0.08, 0.50, 0.60, 0.70, 0.75 |
| | MeOH | 0.01 | 0.01 | 0.01 | 0.01 |
| *Scleroderma citrinum* #2 | EtOAc | 0.50, 0.60, 0.70 | 0.55, 0.65, 0.75 | 0.60, 0.70, 0.75 | 0.60, 0.70, 0.75 |
| | MeOH | 0.90 | 0.50, 0.90 | 0.01 | 0.01 |
| *Scleroderma citrinum* #3 | EtOAc | 0.08, 0.50, 0.60, | 0.08, 0.55, 0.65 | 0.08, 0.60, 0.70 | 0.08, 0.60, 0.70 |
| | MeOH | 0.01 | 0.50 | 0.01, 0.35, 0.80 | 0.01 |
| *Scleroderma citrinum* #4 | EtOAc | 0.60, 0.70 | 0.55, 0.65, 0.75 | 0.60, 0.70, 0.75 | 0.60, 0.70, 0.75 |
| | MeOH | / | / | 0.01 | 0.01 |

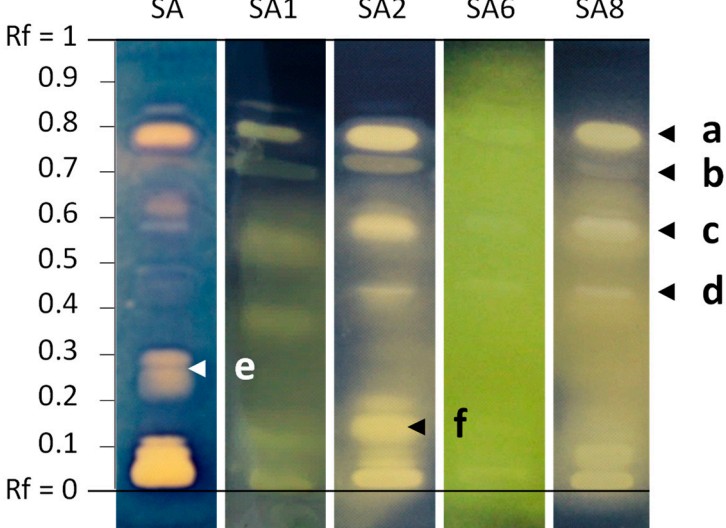

**Figure 3.** TLC bioautograms obtained with the EtOAc extract from *Fomitopsis pinicola* against the wild type strain of *S. aureus* (SA) ATCC 29,213 and the four MDR strains SA1, SA2, SA6, and SA8. The green color observed for SA6 could be due to a poor reduction of MTT. Compounds **a**–**f** are discussed more in details in the text regarding their selectivity.

Regarding *Scleroderma citrinum*, the four samples collected in different times and places displayed variable contents of antibacterial compounds in EtOAc extracts. This could be

due to environmental factors affecting the biosynthesis of secondary bioactive metabolites, or just intrinsic variability within the same species. As depicted in Figure 4, the four extracts were found to contain active compounds **b** and **c**, but the active compound **d** was present only in *S. citrinum* #1 and #3. Similarly, *S. citrinum* #3 was the only extract devoid of compound **a**.

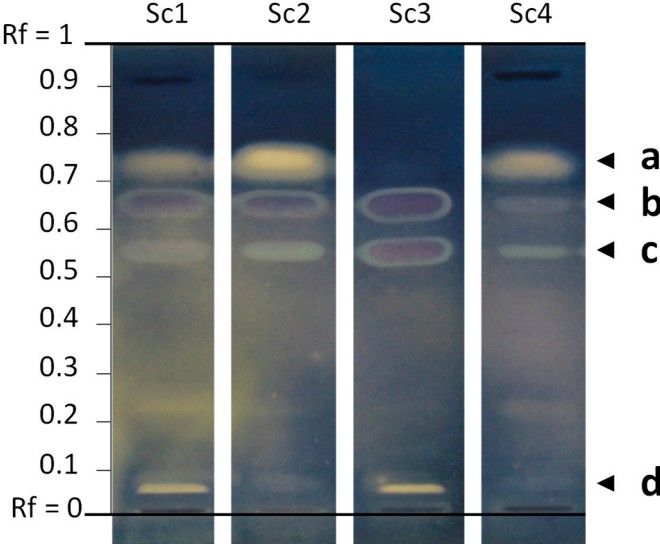

**Figure 4.** TLC bioautograms obtained with the EtOAc extracts from the four *Scleroderma citrinum* samples against the MDR strain SA2. Compounds **a**–**d** are discussed more in details in the text regarding their occurrence and selectivity.

3.3.2. Antibacterial Activity against MDR *E. coli* Strains

Three out of four strains were extended spectrum beta-lactamase (ESBL) producing *E. coli* (EC2, EC5, and EC8). Like MRSA, bacteraemia caused by ESBL producing *E. coli* are associated with higher length of hospital stay and mortality, and are therefore, classified as critical by the WHO. In addition, to beta-lactam antibiotics, these clinical isolates resist to various other commercial drugs; for instance, EC2 and EC5 resist to 15 and 16 antibiotics out of 28 tested, respectively.

As against wild-type strains, EtOAc extracts were found to be more interesting than methanolic extracts when tested against MDR strains (Table 4). Only the methanolic extract from *Fomitopsis pinicola* was still active against *E. coli* clinical isolates: one inhibition band was visible at Rf 0.65 on the bioautogram towards EC5, whereas two less polar molecules inhibited the growth of EC1, one of the most resistant strains in this study; EC6 and EC8 were found to be a little bit more sensitive as four inhibition bands could be observed at Rf between 0.50 and 0.80. Although no clear inhibition zone was detected against the wild-type strain with the methanolic extracts from *Scleroderma citrinum*, a thin but resolved band was visible just above the deposit line for *S. citrinum* #1 and #4 against EC2 and EC8.

**Table 4.** Rf values of inhibition bands induced by mushrooms extracts on bioautograms against the four MDR *E. coli* strains EC2, EC5, EC6, and EC8.

| Mushroom Species | Extract | EC2 | EC5 | EC6 | EC8 |
|---|---|---|---|---|---|
| *Clitocybe nebularis* | EtOAc | 0.78 | 0.78 | 0.78 | 0.78 |
| | MeOH | / | / | / | / |
| *Hypholoma fasciculare* | EtOAc | / | 0.78 | 0.78 | 0.78 |
| | MeOH | / | / | / | / |
| *Pleurotus ostreatus* | EtOAc | 0.78 | 0.78 | 0.40, 0.62 | 0.78 |
| | MeOH | / | / | / | / |

**Table 4.** *Cont.*

| Mushroom Species | Extract | EC2 | EC5 | EC6 | EC8 |
|---|---|---|---|---|---|
| *Lactifluus piperatus* | EtOAc | 0.78 | 0.78 | 0.78 | 0.78 |
| | MeOH | / | / | / | / |
| *Fomitopsis pinicola* | EtOAc | / | 0.55, 0.65, 0.78 | 0.40, 0.55 | 0.55 |
| | MeOH | 0.70, 0.85 | 0.65 | 0.50, 0.60, 0.62, 0.80 | 0.55, 0.65, 0.70, 0.82 |
| *Scleroderma citrinum* #1 | EtOAc | 0.60, 0.70 | 0.01, 0.60 | 0.60, 0.78 | 0.70 |
| | MeOH | 0.01 | / | 0.40 | 0.01 |
| *Scleroderma citrinum* #2 | EtOAc | 0.60, 0.70, 0.75 | 0.01, 0.60, 0.78 | 0.60, 0.78 | 0.60, 0.70, 0.75 |
| | MeOH | 0.90 | / | 0.40 | 0.42 |
| *Scleroderma citrinum* #3 | EtOAc | 0.60, 0.70 | 0.60 | 0.60 | 0.60 |
| | MeOH | 0.40, 0.45 | / | 0.40 | 0.40 |
| *Scleroderma citrinum* #4 | EtOAc | 0.75 | 0.78 | / | 0.75 |
| | MeOH | 0.01 | / | / | 0.01 |

Regarding the EtOAc extracts, inhibition bands were mainly observed in the upper part of bioautograms, indicative of bioactive compounds of relatively low polarity (Figure 5).

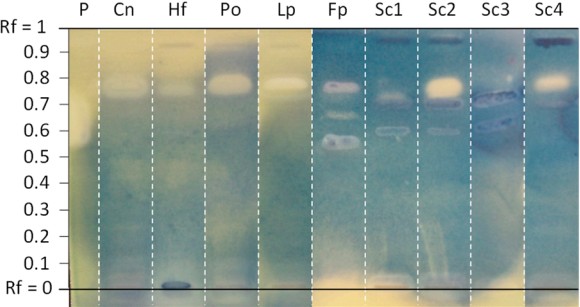

**Figure 5.** TLC bioautograms of polymyxin B (P) and EtOAc extracts (initials of mushroom species) against the MDR *E. coli* strain EC5.

An intense inhibition band at Rf 0.75 was present in several bioautograms obtained with extracts from phylogenetically unrelated mushroom, such as *Pleurotus ostreatus*, *Lactifluus piperatus*, *Fomitopsis pinicola*, or *Scleroderma citrinum*. This inhibition band was also observed in several mushrooms' extracts tested against *S. aureus* MDR strains (Table 3), suggesting that a broad-range antibacterial compound may be produced by most, if not all, fungal lineages. We thus sought to identify the chemical nature of this promising molecule, using the EtOAc extract of *Lactifluus piperatus* as a source, as this compound was predominant in this species.

### 3.4. Identification of Stearic Acid

The EtOAc extract from *Lactifluus piperatus* was first fractionated by centrifugal partition chromatography. The fractions containing the compound of interest were pooled and further separated by semi-preparative HPLC. The compound of interest crystallized as white crystals reflecting visible light. It enhanced the blue fluorescence of primuline on TLC, which is typical of long hydrocarbon chains such as lipids and fatty acids. Its molecular formula was determined to be $C_{18}H_{36}O_2$ on the basis of high-resolution mass spectra, suggesting that this compound could be stearic acid, which was further confirmed by NMR analyses by comparison with a standard.

## 4. Discussion

Bacterial infections still represent a major health problem, amplified by the phenomenon of antibiotic resistance. Micromycetes have long been a reservoir of antibacterial drugs, but higher fungi, i.e., mushroom-forming fungi or macromycetes, are now emerging as a promising source of antimicrobial compounds [9,10].

Generally, antibacterial activity of crude extracts is assessed by microdilution assays; however, this approach provides only a global quantitative assessment of the potential of the extract, but does not bring any information about the number of molecules involved in this antimicrobial activity. Furthermore, results can be affected by antagonist or synergistic effects. For these reasons, we decided to screen extracts by TLC-direct bioautography, which combines preliminary planar chromatographic separation of crude extracts and subsequent application of bacterial suspensions on eluted TLC plates [11]. This strategy enables us to evaluate the number of compounds involved in bioactivity, to assess if different bacterial strains are inhibited by the same or distinct compounds, and to get an idea about the polarity of active compounds according to their elution on the chromatographic plate.

Though we have chosen to order antibacterial activity according to fungal taxonomy (Tables A3–A7), no correlation emerges from our work that would assign any antimicrobial activity to a known fungal lineage. Some chemical structures may be restricted to a limited number of species or genera, but they could equally well have evolved in response to ecological constraints shared by organisms taxonomically unrelated. In fact, since the function of most bioactive compounds from macromycetes is still unknown for their establishment or maintenance in their respective ecological niche, molecules inhibiting bacterial growth in vitro (on TLC plates or in petri dishes) may just have no such function in situ or they may not be subjected to direct selection pressure. Consistent with this view, the distribution of lethal amatoxins among the *Amanita*, *Galerina*, and *Lepiota* genera, shows that (very) active chemical structures have evolved repeatedly in distinct fungal lineages, without an obvious link with ecological features or evolutionary origins [12]. In addition, as experienced here with *Neoboletus erythropus* or *Scleroderma citrinum*, environmental factors such as biotopes or seasons seem to affect both the qualitative and quantitative contents of antibacterial compounds, further blurring any correlation between taxonomy and activity of these molecules. Notwithstanding these open questions about the physiological function of these active metabolites for the fungus, the data obtained here show that most—if not all—mushrooms produce few to several compounds endowed with antibacterial properties, with high or low selectivity, confirming our initial hypothesis.

Our results highlighted that low-polarity metabolites seem to contribute significantly to antibacterial effects. Stearic acid in particular was detected in many samples, yielding an intense inhibition band in the upper part of several bioautograms (Rf around 0.75). This observation was first considered with caution, as it is known that some non-polar products may generate false positives, as hydrophobicity of the compound can act as a repellent for the aqueous bacterial suspension. However, we observed that even more lipophilic fatty acids did not affect the growth of bacteria in our system, corroborating a genuine activity of some low-polarity compounds. It turns out that the antibacterial action of stearic acid and other related fatty acids is well documented [13,14]. It was demonstrated that free fatty acids induce their antibacterial action through several mechanisms, including cell disruption, interference with cellular energy production, enzyme inhibition, and others. The hydrophobicity of fatty acids can be a barrier to drug development; to overcome this drawback, researchers have used liposomes as carriers for stearic acid, and found that this formulation strongly inhibited the growth of multidrug-resistant strains of *S. epidermidis* and *E. faecalis* (MIC of 0.25 µg/mL and 0.5 µg/mL, respectively) [15]. In addition to these fatty acid derivatives, more polar molecules are also involved in the antibacterial arsenal of mushrooms, as evidenced by inhibition bands observed in the bottom part of the bioautograms.

In our screening, the wood-decaying fungus *Fomitopsis pinicola* was the richest in antibacterial compounds, not only against wild-type bacteria, but also against MDR strains.

Both polar and low polarity extracts contained active compounds, which was hardly the case for the other mushroom species. Regardless of bioactivity, the thin-layer chromatographic profile of *Fomitopsis pinicola* extracts revealed an abundance of metabolites, mainly sterol and triterpene derivatives, according to TLC reagents and the literature [15,16]. These compounds probably contribute to the antibacterial activity together with fatty acid derivatives, a hypothesis reinforced by previous reports mentioning the antibacterial activity of triterpenes from polypores [17–19]. *Scleroderma citrinum* is also unveiled here as a promising source of active antibacterial compounds, as it was also found to contain molecules inhibiting the growth of ESBL producing *E. coli* and methicillin-resistant *Staphylococcus aureus*, considered as the most challenging bacterial strains to fight.

Active compounds have now to be purified and identified, and their mechanism of action to be deciphered. Activity against other pathogenic or MDR bacterial strains could be also evaluated to assess the selectivity of these fungal compounds, and their potential for further drug candidate development.

**Author Contributions:** Conceptualization, M.B. and A.U.; mushroom harvesting, M.B. and C.H.; mushroom identification, J.-M.B.; data curation, J.-M.B. and A.U.; formal analysis, C.H. and J.-M.B.; investigation, C.H.; methodology, C.H., M.B. and A.U.; project administration, M.B. and A.U.; resources, G.P.; supervision, M.B., G.P. and A.U.; writing—original draft, C.H., J.-M.B. and A.U.; writing—review and editing, M.B. and G.P. All authors have read and agreed to the published version of the manuscript.

**Funding:** This research received no external funding.

**Institutional Review Board Statement:** Not applicable.

**Informed Consent Statement:** Not applicable.

**Data Availability Statement:** The data presented in this study are available on request from the corresponding author.

**Acknowledgments:** The authors are grateful to the Société Mycologique de Strasbourg, and particularly to J.-F. Stambach and R. Wiest, for their help with preliminary mushroom identification. D. Keller from UMR 7290, University of Strasbourg, is gratefully acknowledged for cultivation of MDR strains.

**Conflicts of Interest:** The authors declare no conflict of interest.

## Appendix A

**Table A1.** Antibiograms of the four *Staphylococcus aureus* multidrug resistant strains.

| Antibiotic | Class of Antibiotics | SA1 | SA2 | SA6 | SA8 |
|---|---|---|---|---|---|
| Oxacillin | Penicillins | S | R | R | R |
| Ceftobiprol | Cephalosporin-5G | S | S | S | S |
| Amikacin | Aminoglycosides | R | R | R | R |
| Gentamicin | Aminoglycosides | S | S | S | S |
| Tobramycin | Aminoglycosides | S | R | R | S |
| Ofloxacin | Fluoroquinolones | S | R | R | S |
| Levofloxacin | Fluoroquinolones | S | R | R | S |
| Erythromycin | Macrolides | R | S | S | R |
| Clindamycin | Lincosamides | S | S | ND | R |
| Pristinamycin | Streptogramins | S | S | S | S |
| Tigecycline | Glycylcyclines | S | S | S | S |
| Trimethoprim/sulfamethoxazole | Sulfamide/diaminopyrimidine | S | S | S | S |
| Fosfomycin | Phosphonic acids | S | S | S | S |
| Linezolid | Oxazolidinones | S | S | S | S |
| Rifampicin | Ansamycins | S | S | S | S |
| Fusidic acid | Fusidanes | R | S | R | S |
| Vancomycin | Glycopeptide | ND | ND | S | ND |
| Teicoplanin | Glycopeptide | ND | ND | S | ND |

R: resistant; S: sensitive; ND: Not determined.

**Table A2.** Antibiograms of the four *Escherichia coli* multidrug resistant strains.

| Antibiotic | Class of Antibiotics | EC2 | EC5 | EC6 | EC8 |
|---|---|---|---|---|---|
| Amoxicillin | Aminopenicillins | R | R | R | R |
| Amoxicillin/clavulanic acid | Aminopenicillins/β-lactamase inhibitor | R | R | S | R |
| Mecillinam | Amidinopenicillins | S | S | S | R |
| Ticarcillin | Carboxypenicillins | R | R | R | ND |
| Ticarcillin/clavulanic acid | Carboxypenicillins/β-lactamase inhibitor | R | ND | S | ND |
| Temocillin | Carboxypenicillins | ND | ND | ND | ND |
| Piperacillin | Ureidopenicillins | R | R | S | ND |
| Piperacillin/tazobactam | Ureidopenicillins/β-lactamase inhibitor | S | S | S | R |
| Cefoxitin | Cephamycins | S | S | S | S |
| Cefixime | Cephalosporin-3G | ND | R | S | S |
| Ceftriaxone | Cephalosporin-3G | ND | R | S | S |
| Cefotaxime | Cephalosporin-3G | R | ND | ND | ND |
| Ceftazidime | Cephalosporin-3G | R | R | S | S |
| Cefepime | Cephalosporin-4G | R | R | ND | ND |
| Aztreonam | Monobactams | R | R | ND | ND |
| Ertapenem | Carbapenems | S | S | S | S |
| Imipenem | Carbapenems | S | ND | ND | ND |
| Meropenem | Carbapenems | S | S | ND | ND |
| Amikacin | Aminoglycosides | S | S | S | S |
| Gentamicin | Aminoglycosides | R | R | R | R |
| Tobramycin | Aminoglycosides | R | R | ND | ND |
| Nalidixic acid | Quinolones | ND | R | S | R |
| Ciprofloxacin | Fluoroquinolones | R | S | R | R |
| Levofloxacin | Fluoroquinolones | R | R | ND | ND |
| Ofloxacin | Fluoroquinolones | R | R | S | R |
| Trimethoprim/sulfamethoxazole | Sulfamide/diaminopyrimidine | R | R | R | S |
| Fosfomycin | Phosphonic acids | S | R | S | S |
| Nitrofurantoin | Nitrofurans | S | S | S | S |

R: resistant; S: sensitive; ND: Not determined.

**Table A3.** Antibacterial activity induced by Agaricales mushrooms against wild-type bacteria expressed as Rf values of inhibition bands.

| Mushroom Species | Extract | *E. faecalis* Rf Values | *S. aureus* Rf Values | *S. epidermidis* Rf Values | *E. coli* Rf Values | *P. aeruginosa* Rf Values |
|---|---|---|---|---|---|---|
| *Agaricus bisporus* | EtOAc | 0.45, 0.81 | 0.41, 0.59, 0.75 | 0.49, 0.81 | 0.06, 0.43, 0.73–0.78 | / |
| | MeOH | 0, 0.08, 0.15, 0.75, 0.85 | 0.85 | 0.88 | 0.15, 0.85 | / |
| *Amanita citrina* | EtOAc | 0.28, 0.45, 0.78, 0.80–0.84 | 0, 0.40–0.44, 0.70–0.94 | 0.10, 0.49, 0.79–0.84 | 0.25–0.28, 0.41–0.45, 0.86 | 0.78–0.83 |
| | MeOH | 0.08, 0.85 | 0.85 | 0.92 | 0.02, 0.70 | / |
| *Amanita muscaria* | EtOAc | 0.09, 0.26, 0.37, 0.43, 0.54, 0.58–0.63, 0.75, 0.78–0.83 | 0.04, 0.08, 0.10, 0.63, 0.68, 0.72–0.76, 0.77–0.80 | 0.03, 0.73 | 0–0.10, 0.76 | 0.70 |
| | MeOH | 0.80, 0.86 | 0, 0.48 | 0.11, 0.59 | 0.70 | / |
| *Armillaria cepistipes* var. *pseudobulbosa* | EtOAc | 0.09, 0.26, 0.37, 0.43, 0.54, 0.58–0.63, 0.75, 0.78–0.83 | 0.04, 0.08, 0.10, 0.63, 0.68, 0.72–0.76, 0.77–0.80 | 0.03, 0.73 | 0–0.10, 0.76 | 0.70 |
| | MeOH | 0.80, 0.86 | 0, 0.48 | 0.11, 0.59 | 0.70 | / |
| *Armillaria ostoyae* | EtOAc | 0.79 | 0.06, 0.09, 0.15, 0.78 | 0.09, 0.80 | 0.04, 0.10, 0.75 | / |
| | MeOH | / | / | 0 | / | / |

**Table A3.** *Cont.*

| Mushroom Species | Extract | *E. faecalis* Rf Values | *S. aureus* Rf Values | *S. epidermidis* Rf Values | *E. coli* Rf Values | *P. aeruginosa* Rf Values |
|---|---|---|---|---|---|---|
| *Bolbitius variicolor* | EtOAc | 0.47–0.50, 0.84–0.87 | 0.78 | 0.75–0.83 | 0.69 | / |
| | MeOH | 0, 0.26, 0.66–0.73 | 0 | 0, 0.80 | 0.03 | / |
| *Clitocybe nebularis* | EtOAc | 0.05, 0.75–0.80 | 0.06, 0.15, 0.40, 0,70–0.78 | 0.04, 0.06, 0.10, 0.21, 0.48, 0.65, 0.76–0.84 | 0.03, 0.04, 0.06, 0.18, 0.25, 0.43, 0.73–0.80 | 0.16, 0.79 |
| | MeOH | / | / | / | / | / |
| *Cortinarius semisanguineus* | EtOAc | 0.63–0.69 | 0–0.05, 0.31, 0.85–0.94 | 0–0.05, 0.31, 0.85–0.94 | / | / |
| | MeOH | / | / | 0.11, 0.58 | / | / |
| *Gymnopus luxurians* | EtOAc | 0.03, 0.30, 0.50, 0.73–0.78 | / | / | 0.75–0.85 | 0.75–0.81 |
| | MeOH | / | / | 0.13, 0.60 | / | / |
| *Hypholoma fasciculare* | EtOAc | 0–0.05, 0.48–0.52, 0.73, 0.84–0.87 | 0–0.05, 0.75–0.79 | 0–0.05, 0.60, 0.74 | 0.05, 0.70 | 0 |
| | MeOH | 0.04, 0.05, 0.10, 0.34, 0.38 | 0.05, 0.29, 0.39 | 0.20 | / | 0.04, 0.21, 0.31–0.35, 0.41 |
| *Macrolepiota procera* | EtOAc | 0.03, 0.06, 0.49, 0.56, 0.80–0.84 | 0.75–0,79, 0.83–0.89 | / | / | / |
| | MeOH | / | / | 0.11, 0.59 | / | / |
| *Mycena rosea* | EtOAc | 0.04, 0.55, 0.79, 0.81 | 0.04, 0.06, 0.15, 0.70–0.79 | 0.05, 0.10, 0.21, 0.78–0.84 | 0.25, 0.73–0.80 | 0.16, 0.79 |
| | MeOH | / | / | / | / | / |
| *Parasola auricoma* | EtOAc | 0.09, 0.28, 0.79, 0.81 | 0.05, 0.08, 0.15, 0.41, 0.71–0.94 | 0.21, 0.79–0.83 | 0.44, 0.73–0.78 | 0.79 |
| | MeOH | / | / | 0 | / | / |
| *Pleurotus ostreatus* | EtOAc | 0.65, 0.78–0.83 | 0.06, 0.15, 0.59, 0.70–0.79 | 0.06, 0.10, 0.13, 0.21, 0.68, 0.79–0.84 | 0.41, 0.63, 0.74–0.80 | 0.79 |
| | MeOH | / | / | 0 | / | / |
| *Volvopluteus gloiocephalus* | EtOAc | 0.03, 0.08, 0.16, 0.29, 0.38, 0.41, 0.49–0.50, 0.59, 0.64, 0.71, 0.81–0.83 | 0.04, 0.15, 0.55, 0.64–0.66, 0.73–0.75, 0.76–0.81 | 0.61, 0.74, 0.76 | 0.13, 0.43–0.50, 0.69–0.80 | 0.05, 0.76–0.80 |
| | MeOH | 0 | / | 0 | / | / |

**Table A4.** Antibacterial activity induced by Boletales mushrooms against wild-type bacteria expressed as Rf values of inhibition bands.

| Mushroom Species | Extract | *E. faecalis* Rf Values | *S. aureus* Rf Values | *S. epidermidis* Rf Values | *E. coli* Rf Values | *P. aeruginosa* Rf Values |
|---|---|---|---|---|---|---|
| *Hygrophoropsis aurantiaca* | EtOAc | 0.03, 0.08, 0.10, 0.45, 0.79–0.81 | 0.80 | 0.65, 0.80 | 0.03, 0.76 | / |
| | MeOH | 0.59 | / | 0, 0.56 | / | / |
| *Neoboletus erythropus* #1 | EtOAc | 0, 0.14, 0.41–0.45, 0.53, 0.65, 0.79–0.83 | 0.73–0.76, 0.77–0.81 | 0.63, 0.70 | 0.66, 0.78 | / |
| | MeOH | / | / | / | / | / |



**Table A4.** *Cont.*

| Mushroom Species | Extract | *E. faecalis* Rf Values | *S. aureus* Rf Values | *S. epidermidis* Rf Values | *E. coli* Rf Values | *P. aeruginosa* Rf Values |
|---|---|---|---|---|---|---|
| *Neoboletus erythropus* **#2** | EtOAc | 0.01, 0.10, 0.42, 0.69–0.81 | 0.04, 0.06, 0.45–0.48, 0.60–0.65, 0.69–0.75, 0.89 | 0.04, 0.06, 0.45–0.48, 0.60–0.65, 0.69–0.75, 0.86–0.89 | 0.75–0.79 | 0–0.03, 0.75–0.80 |
| | MeOH | / | 0–0.04, 0.70, 0.93 | 0.44 | / | / |
| *Paxillus involutus* | EtOAc | 0.03–0.10, 0.15, 0.20, 0.30–0.33, 0.48–0.51, 0.54–0.61, 0.65, 0.71, 0.79–0.86 | 0.04, 0.69–0.73, 0.75–0.79 | 0.79 | 0.80 | 0.70 |
| | MeOH | / | / | 0.90 | / | / |
| *Scleroderma citrinum* **#1** | EtOAc | 0.01, 0.03, 0.04–0.05, 0.15, 0.48–0.50, 0.60–0.63, 0.71, 0.75, 0.79–0.84 | 0, 0.02, 0.04, 0.05, 0.74, 0.79 | 0.05, 0.64, 0.78 | 0.61, 0.71 | 0.05, 0.11, 0.25, 0.43 |
| | MeOH | / | / | 0.89 | / | / |
| *Scleroderma citrinum* **#2** | EtOAc | 0.05–0.08, 0.44, 0.57, 0.63–0.66, 0.69, 0.73, 0.78–0.83 | 0.02, 0.05, 0.56–0.61, 0.65–0.68, 0.71–0.73, 0.76–0.81 | 0.63, 0.68–0.77 | 0.63, 0.69–0.78 | 0.60, 0.69 |
| | MeOH | / | / | 0.89 | / | / |
| *Scleroderma citrinum* **#3** | EtOAc | 0.05, 0.41, 0.49, 0.70 | 0.02, 0.43, 0.61, 0.79 | 0.03 | 0.45, 0.63, 0.71 | / |
| | MeOH | / | 0, 0.45, 0.73, 0.77, 0.94 | 0 | 0.13 | / |

**Table A5.** Antibacterial activity induced by Pezizales mushrooms against wild-type bacteria expressed as Rf values of inhibition bands.

| Mushroom Species | Extract | *E. faecalis* Rf Values | *S. aureus* Rf Values | *S. epidermidis* Rf Values | *E. coli* Rf Values | *P. aeruginosa* Rf Values |
|---|---|---|---|---|---|---|
| *Morchella esculenta* **#1** | EtOAc | 0.80 | 0.65, 0.77 | 0.84 | / | 0.70 |
| | MeOH | 0.65 | 0, 0.30, 0.55–0.56, 0.73 | / | 0.56 | / |
| *Morchella esculenta* **#2** | EtOAc | 0.03, 0.14, 0.22, 0.78 | / | 0.04, 0.06, 0.10, 0.23 | 0.04, 0.16 | 0.09, 0.19 |
| | MeOH | / | 0.69 | / | / | / |
| *Peziza vesiculosa* | EtOAc | 0.43, 0.51, 0.58, 0.63, 0.78–0.83 | 0.73–0.76, 0.77–0.81 | 0.74 | 0.78 | / |
| | MeOH | 0, 0.64–0.72 | 0, 0.59, 0.81, 0.94 | 0, 0.80 | 0 | 0, 0.88 |

**Table A6.** Antibacterial activity induced by Polyporales mushrooms against wild-type bacteria expressed as Rf values of inhibition bands.

| Mushroom Species | Extract | *E. faecalis* Rf Values | *S. aureus* Rf Values | *S. epidermidis* Rf Values | *E. coli* RF Values | *P. aeruginosa* Rf Values |
|---|---|---|---|---|---|---|
| *Fomitopsis pinicola* | EtOAc | 0.01, 0.03, 0.04, 0.08, 0.15, 0.21, 0.30, 0.37, 0.41, 0.53, 0.56, 0.62, 0.66, 0.71, 0.82, 0.89 | 0–0.05, 0.08, 0.21–0.26, 0.30, 0.48, 0.58, 0.64, 0.73, 0.75–0.79 | 0.04, 0.05, 0.06, 0.41, 0.53, 0.58, 0.63, 0.73–0.76, 0.83 | 0.03, 0.44, 0.64, 0.74 | 0.45, 0.55, 0.63, 0.71–0.74 |
| | MeOH | 0.31, 0.34, 0.38, 0.41, 0.50, 0.56, 0.60, 0.75, 0.81 | 0.28–0.35, 0.38, 0.45, 0.60–0.73, 0.76 | 0.28, 0.33, 0.40, 0.53, 0.56, 0.70, 0.71–0.76, 0.80 | 0.24, 0.27, 0.33, 0.39, 0.48, 0.57, 0.62 | 0.03, 0.08, 0.33–0.38, 0.40–0.45, 0.46–0.49, 0.51, 0.54–0.58, 0.59, 0.63, 0.66 |
| *Ganoderma applanatum* | EtOAc | 0.19, 0.29, 0.58, 0.82–0.86 | 0.28, 0.54, 0.76–0.81 | 0.75 | 0.72 | / |
| | MeOH | 0–0.05 | 0–0.06, 0.63, 0.75–0.81, 0.85–0.90 | 0 | 0.06, 0.23 | / |
| *Meripilus giganteus* | EtOAc | 0.03, 0.08, 0.19, 0.45, 0.55, 0.66, 0.80–0.83 | 0, 0.05, 0.55, 0.81 | 0.63, 0.74 | 0.03, 0.74 | / |
| | MeOH | / | / | 0 | 0 | / |
| *Polyporus umbellatus* | EtOAc | 0.03, 0.08, 0.18, 0.28, 0.43–0.45, 0.60, 0.68, 0.79–0.83 | 0.05, 0.68, 0.81 | 0.18, 0.25, 0.43, 0.81 | 0.03, 0.43, 0.71–0.81 | / |
| | MeOH | 0.75, 0.78 | 0 | 0.11, 0.59 | / | / |
| *Trametes versicolor* | EtOAc | 0.28, 0.79–0.84 | 0.16, 0.43, 0.60–0.69, 0.75–0.84 | 0.18, 0.21, 0.66, 0.81 | 0.31, 0.73–0.79 | / |
| | MeOH | / | 0.18 | 0.24 | / | / |

**Table A7.** Antibacterial activity induced by Russulales mushrooms against wild-type bacteria expressed as Rf values of inhibition bands.

| Mushroom Species | Extract | *E. faecalis* Rf Values | *S. aureus* Rf Values | *S. epidermidis* Rf Values | *E. coli* Rf Values | *P. aeruginosa* Rf Values |
|---|---|---|---|---|---|---|
| *Lactarius helvus* | EtOAc | 0.04, 0.22, 0.41, 0.75–0.84 | 0.06, 0.15, 0.22, 0.40, 0.61–0.66, 0.75–0.83 | 0.06, 0.77, 0.83–0.89 | 0.01, 0.04, 0.08, 0.24, 0.41, 0.69 | 0.66–0.70, 0.80–0.84 |
| | MeOH | / | 0 | 0 | 0 | / |
| *Lactifluus piperatus* | EtOAc | 0.05, 0.43, 0.80–0.85 | 0.77–0.83, 0.86 | 0.25, 0.40, 0.76 | 0.48, 0.81 | 0.75–0.76 |
| | MeOH | 0.03, 0.79 | / | / | / | / |
| *Russula integra* | EtOAc | 0.03, 0.06, 0.08, 0.16, 0.28, 0.44–0.46, 0, 80–0.86 | 0.76–0.81 | 0.25, 0.40, 0.76 | 0.80 | 0.79 |
| | MeOH | 0, 0.86 | / | 0–0.05, 0.79–0.84 | / | / |
| *Russula lepida* | EtOAc | 0, 0.16–0.20, 0.29, 0.68, 0.79–0.83 | 0, 0.05, 0.08, 0.19, 0.60, 0.66, 0.70, 0.79–0.85 | 0.81–0.84 | 0.04, 0.19, 0.54, 0.60, 0.78 | 0.19, 0.81–0.86 |
| | MeOH | / | / | 0, 0.68 | / | / |
| *Scutiger pes-caprae* | EtOAc | 0.03, 0.07, 0.15, 0.31, 0.50, 0.68, 0.80–0.86 | 0.75, 0.80–0.85 | 0.05, 0.64, 0.76–0.81 | 0.80 | / |
| | MeOH | 0.01 | 0 | 0–0.06 | 0.95 | / |

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
