# Peer review of "Screening for Antibacterial Activity of French Mushrooms against Pathogenic and Multidrug Resistant Bacteria"

_applsci, doi:10.3390/app12105229_

Round 1

Reviewer 1 Report

1. Figure 3, Page 10 = For SA6 strain, the color is green and bands are hardly visible. Authors have acknowledged this. Do authors see the same trend for other mushroom extracts as well. 

2. Mushroom compounds run over TLC bioautograms based on their polarity. Modifications like oxidation during sample preparation can change the polarity of these compounds and elute at separate time. These modified compounds may or may not loose their effectiveness against bacteria. Its also possible that bands at different Rf value belong to different modified compounds. Have authors thought about these scenarios and are there any evidences that they could occur in this study. 

3. It will be great if authors could clarify why EtOAc extracts are more effective against bacteria than MeOH extracts.

4. Are EtOAc and MeOH by themselves active against bacteria? Will it be beneficial to have a control where authors prepare TLC autograms with these compounds against bacteria in figure 1?

5. There is a grammar mistake in line 262 "from a same species". 

Author Response

Thank you for the careful attention given to our manuscript. Please find hereafter our answers and comments to specific points you have raised:

  1. Figure 3, Page 10 = For SA6 strain, the color is green and bands are hardly visible. Authors have acknowledged this. Do authors see the same trend for other mushroom extracts as well. 

R1: Actually, we have rarely encountered this green staining problem, and recently discovered that it was probably related to a poor-quality batch of MTT. Even though contrast is not so clear on the picture, it was more obvious to the eye.

  1. Mushroom compounds run over TLC bioautograms based on their polarity. Modifications like oxidation during sample preparation can change the polarity of these compounds and elute at separate time. These modified compounds may or may not loose their effectiveness against bacteria. Its also possible that bands at different Rf value belong to different modified compounds. Have authors thought about these scenarios and are there any evidences that they could occur in this study. 

R2: When extracting and purifying natural products, there is always this questioning: is it a native product or an extraction artefact? In any case, sensitive compounds (to oxidation or something else) are not suitable for industrial development. So we consider that these compounds are of interest for drug development, even if they could be artefacts.

  1. It will be great if authors could clarify why EtOAc extracts are more effective against bacteria than MeOH extracts.

R3: At this point of the project, we cannot explain why low-polarity extracts are more active than methanolic ones. This will require the purification and structural characterization of every bioactive compound, currently in progress, and then elucidation of mechanism of action.

  1. Are EtOAc and MeOH by themselves active against bacteria? Will it be beneficial to have a control where authors prepare TLC autograms with these compounds against bacteria in figure 1?

R4: When bioautography is performed, there is absolutely no trace of organic solvents on the TLC plate, neither from the extraction solvent nor the elution solvent, as plates are carefully dried. Therefore, the effect of MeOH or EtOAc cannot be taken into account in these bioassays.

  1. There is a grammar mistake in line 262 "from a same species". 

R5: sentence modified.

Reviewer 2 Report

This paper is ineresting and suitable for the publication

Author Response

Thank you for your comment and the careful attention given to our manuscript.

Reviewer 3 Report

Good manuscript.

Author Response

(The authors gave the same response as above.)

Reviewer 4 Report

Authors of the reviewed article evaluated the antibacterial activity of mushroom extracts. For study, they choose 31 fungal species growing in France. Extracts were evaluated against pathogenic wild type and some MDR strains of bacteria. To determine biological activity thin-layer chromatography-direct bioautography (TLC-DB) was implemented. In my opinion, the presented article is based on good ideas because searching for new active (antibacterial or antifungal) substances is very important to challenge for sciences.

The manuscript is well written, the objectives are well formulated. However, please clarify or completed the following points:

- In methodology, especially in point Extract preparation should be given the mass of dry material used for extraction. To compare the presence or concentration of active substances between the different mushroom extracts, the mass of these samples should be similar.

- Did Authors try to perform the two-dimensional TLC of fungal extracts, to be certain that all chemicals were separated?

- In line 157 Lactifluus piperatus should be written in italics

- Authors write “Regarding Scleroderma citrinum, the four samples collected in different times and places displayed variable contents of antibacterial compounds in EtOAc extracts, illustrating the influence of environmental factors on the biosynthesis of bioactive compounds.” Is it possible to relate the environmental conditions of mushroom growth with their chemical composition?

- Did Authors try to identify other active substances (other than stearic acid) present in extracts? This is the key point to use these substances as drug candidates.

Author Response

Thank you for the careful attention given to our manuscript. Please find hereafter our answers and comments to specific points you have raised:

  1. In methodology, especially in point Extract preparationshould be given the mass of dry material used for extraction. To compare the presence or concentration of active substances between the different mushroom extracts, the mass of these samples should be similar.

R1: The masses differed according to the samples harvested, but extractions were of course performed in the same conditions, especially with the same ratio of solvent. This point has been clarified in the revised version.

  1. Did Authors try to perform the two-dimensional TLC of fungal extracts, to be certain that all chemicals were separated?

R2: Due to the high number of data generated for this study (more than 420 bioautograms), one-dimensional chromatography was more suitable. However, this is very relevant suggestion for the further analysis of the more promising extracts !

  1. In line 157 Lactifluus piperatus should be written in italics

R3: done.

  1. Authors write “Regarding Scleroderma citrinum, the four samples collected in different times and places displayed variable contents of antibacterial compounds in EtOAc extracts, illustrating the influence of environmental factors on the biosynthesis of bioactive compounds.” Is it possible to relate the environmental conditions of mushroom growth with their chemical composition?

R4: It is true that it is hard to relate for sure chemical composition with extrinsic factors. It is known that external parameters can affect the biosynthesis of secondary metabolites, but variations can also be due to intrinsic variability. This sentence has been slightly modified accordingly, as well as the sentence line 262.

  1. Did Authors try to identify other active substances (other than stearic acid) present in extracts? This is the key point to use these substances as drug candidates.

R5: The purification and identification of other active substances is currently in progress.